# Hsa_circ_0044301 Regulates Gastric Cancer Cell’s Proliferation, Migration, and Invasion by Modulating the Hsa-miR-188-5p/DAXX Axis and MAPK Pathway

**DOI:** 10.3390/cancers14174183

**Published:** 2022-08-29

**Authors:** Fei Jiang, Guangxi Liu, Xiaowei Chen, Qiong Li, Fujin Fang, Xiaobing Shen

**Affiliations:** 1Key Laboratory of Environmental Medical Engineering and Education Ministry, Nanjing Public Health College, Southeast University, Nanjing 210009, China; 2Department of Epidemiology and Health Statistics, School of Public Health, Southeast University, Nanjing 210009, China; 3Department of Occupational and Environmental Health, School of Public Health, Southeast University, Nanjing 210009, China

**Keywords:** gastric cancer, circRNA, siRNA, miRNA, ERK1/2

## Abstract

**Simple Summary:**

This study aimed to investigate whether circRNA could be potential prognosis or therapeutic target. And we found the upregulated hsa_circ_0044301 was positively correlated with the 5-year survival rate of patients, which also could influence the proliferation, migration and invasion of gastric cancer cells in vitro and in vivo. Mechanically, it could act as the sponge of hsa-miR-188-5p and regulate the expression and function of targeted gene DAXX. In addition, this circRNA could also modulate the effect of GDC-0994 on ERK1/2 or 5-FU in cells. These findings have made a significant contribution to the study of circRNA in the treatment field of gastric cancer. Meanwhile this is the first detailed investigation of hsa_circ_0044301 in gastric cancer, and the circRNA has the value of further animal and clinical translation.

**Abstract:**

**Background:** Despite advances in diagnostic and therapeutic technologies, the prognosis of patients with gastric cancer (GC) remains poor, necessitating further search for more effective therapeutic targets and markers for prognosis prediction. Circular RNA (circRNA) plays a role in various diseases, including GC. **Methods:** CircRNA expression in GC tissues was detected by circRNA microarray and quantitative reverse transcription polymerase chain reaction (qRT-PCR). The correlation between circRNA-0044301 and patient survival was analyzed by log-rank test and Cox regression analysis. Next, in vitro characterization and functional analysis of circRNA-0044301 was done by various assays using RNase R, actinomycin D, and RNA fluorescence in situ hybridization, as well as investigations into its use as a drug to treat tumors in a subcutaneous tumorigenesis model. RNA immunoprecipitation and dual-luciferase reporter assays were used to identify circRNA-0044301-related miRNA (miRNA-188-5p), key proteins of the related pathway (ERK1/2), and the downstream target DAXX. Finally, we investigated the relationship between circRNA-0044301 and ravoxertinib (GDC-0994) and 5-fluorouracil (5-FU) using qRT-PCR, Western blotting, and CCK8 assays. **Results:** CircRNA-0044301 was upregulated in tissues and cancer cells compared to its levels in controls, related to patient prognosis, and its specific siRNA-vivo could slow tumor growth. On the mechanism, it acted as a sponge of miRNA-188-5p, could regulate the downstream target DAXX, and modulated the effect of GDC-0994 on ERK1/2 and 5-FU in cells. **Conclusions:** CircRNA-0044301/miRNA-188-5p/DAXX (ERK1/2) may be a key axis in GC progression, and circRNA-0044301 has immense potential to be a therapeutic target for GC.

## 1. Background

Gastric cancer (GC) is an aggressive tumor that ranks fifth in incidence and third in mortality based on 2018 GLOBLE cancer statistics [1]. However, occurrence of GC is not evenly distributed worldwide, and more than half of all cases are reported from Eastern Asia, Central and Eastern Europe, and Central and South America [2]. Although diagnostic and therapeutic technologies have been developing rapidly, this disease still poses a great threat to human health. Patients with GC have a poor prognosis [3,4,5], with an estimated five-year survival rate of 10% for patients with advanced disease [6]. The main reason for this dismal outlook may be the lack of biomarkers with high specificity, sensitivity, and measures to allow early detection and therapy of GC [7]. Therefore, clarifying the underlying molecular mechanisms to discover potential key molecules, axes, or pathways is important.

Circular RNA (circRNA) is a new class of molecules that has gained increasing attention from researchers. Stability, abundance, and expression of circRNA is specific for tissues, cells, and disease progression [8,9,10]. While circRNA was initially thought to be errant byproducts of splicing with no function [11,12,13], with the development and improvement of high-throughput sequencing technology and experimental technology [14], circRNA was found to be functional. It exists widely in humans [8] as well as plants [15], and is implicated in various diseases [16], including GC.

Current research into the role of circRNA in disease has mainly focused on circRNA as miRNA sponges that adsorb miRNA and modulate the role of downstream target genes. For example, the circRNA_0005075/miR-431/p53/EMT axis inhibits GC growth and metastasis [17]. CircPVT1 can promote GC cell proliferation through the circPVT1/miR-125 axis, and its level is an independent prognostic indicator for the overall survival and disease-free survival of GC patients [18]. The circLARP4/miR-424/LATS1 axis regulates GC cell proliferation and invasion [19], and research suggests that the role of circRNA as miRNA molecular sponges is conserved in different species and is not a rare phenomenon.

In this study, we explored the role of upregulated hsa_circ_0044301 (circRNA-0044301) in vitro and in vivo and found that this circRNA could decrease the tumor volume when it was knocked down, which also could act as the sponge of hsa-miR-188-5p (miRNA-188-5p) and related with targeted gene DAXX. In addition, circRNA-0044301 could also modulate the effect of GDC-0994 on ERK1/2 or 5-FU in cells. To our knowledge, this is the first detailed study on the role and related mechanisms of circRNA-0044301 in GC, and it has therapeutic potential in treatment of GC.

## 2. Methods

### 2.1. Tissues

A total of 61 pairs of GC tissues and adjacent normal tissues were collected for this study. After samples were obtained, they were transferred into the liquid nitrogen tank for preservation. The ethical application for this study was approved by the Ethics Committee of Nanjing First Hospital.

Inclusion criteria: (1) Patients with postoperative pathological diagnosis as GC; (2) have not received chemoradiotherapy or other surgery before the initial GC surgery.

Exclusion criteria: (1) Have received chemoradiation or other surgical treatment; (2) GC patients with other tumors or other digestive system diseases.

### 2.2. Cells

We purchased the human GC cell lines MKN-28 (MKN-74) and MKN-45 and the human normal epithelial cell line GES-1 from Cellcook Biotech Co., Ltd. (Guangzhou, China) with the short tandem report (STR) information report. The GC lines AGS, HGC-27, and MGC-803 were purchased from the Chinese Academy of Sciences Cell Library (Shanghai, China) with the STR information report by Applied Biosystems. The culture medium for all cell lines was Roswell Park Memorial Institute (RPMI), 10% fetal bovine serum (FBS), and 1% penicillin and streptomycin (Gibco Life Technologies, Carlsbad, CA, USA). With the exception of MKN-45 cells, which grow in a semi-affixed wall and semi-suspended pattern, all other cell lines exhibit adherent growth patterns. All cells were cultured with 5% CO_2_ at 37 °C.

### 2.3. Quantitative Reverse Transcription Polymerase Chain Reaction (qRT-PCR)

TRIzol reagent was used to isolate the total RNA from tissues and cells (GenStar, Nanjing, China). Reverse transcription kits were used to synthesize the cDNA of isolated RNA (GenStar, Nanjing, China; Takara Bio, Japan, and RIBO Bio, Guangzhou, China). Then, SYBR Green PCR Kit was performed to quantify circRNA, mRNA, and miRNA (Yeasen Biotech Co., Ltd., Shanghai, China). The sequence of circRNA, miRNA, and mRNA was designed and synthesized by GenePharma (Shanghai, China), Generay Biotech Co. Ltd. (Nanjing, China) and Shanghai sangon Co. Ltd. (Shangai, China), respectively. GAPDH and U6 were used as the reference. The relevant primer information is listed in Appendix A.

### 2.4. Sanger Sequencing

To further verify the specificity and accuracy of circRNA primer amplification production, Sanger sequencing was performed by Qingke Biotechnology (Nanjing, China). 

### 2.5. Vector Construction and Cell Transfection

To knock down circRNA-0044301, small inhibitory RNA (siRNA) targeting the back-splice junction site of circRNA-0044301 or a negative control was synthesized (Hanheng, Shanghai, China) that was named si-circRNA-0044301 or nc-circRNA-0044301, respectively. miRNA-188-5p mimics were built by Hanheng (Shanghai, China). DAXX siRNA was synthesized by Generay Biotech Co. Ltd. (Nanjing, China). The mimics and siRNA were transiently transfected into GC cells with special transfection reagent (RIBO Bio, Guangzhou, China). The dose used in this research (50 nM) was in accordance with the manufacturer’s protocols. The sequences of circRNA-0044301 siRNA, miRNA-188-5p mimics, and DAXX siRNA and negative control are set out in Appendix A.

### 2.6. RNA Fluorescence In Situ Hybridization (FISH)

The circRNA-0044301 oligonucleotide-modified probe sequence was synthesized by Ribo Bio Technology Co Ltd. (Guangzhou, China). After the cells (HGC-27 and MKN-28) grew to logarithmic stage, the cells were counted and planted on the 24-well plate. When the degree of cell fusion reached 60–70%, the cells were cleaned with PBS (phosphate buffer saline), fixed with 4% paraformaldehyde, permeated with PBS containing 0.5% Triton X-100, treated with hybridization solution, stained, and photographed with microscope (Zeiss, Jena, Germany).

### 2.7. Actinomycin D

Cultured HGC-27 and MKN-28 cells in the logarithmic growth phase were seeded at 15 × 10^4^ cells into a 24-well plate. The cells were treated with actinomycin D at 1 µM after 24 h. The cells were then collected and circRNA-0044301 and linear RNA expression were detected after 0 h, 4 h, 8 h, 12 h, and 24 h to explore the half-life variations of circRNA-0044301 and linear RNA.

### 2.8. RNase R Assay

After extracting the total cell RNA using the TRIzol method, the concentration of RNA was measured. Then, according to the protocol, the total RNA was treated with a specific concentration of RNase R and the expression of circRNA-0044301 and linear RNA was detected using qRT-PCR to explore the stability variations of circRNA-0044301 and linear RNA.

### 2.9. Cell Counting Kit-8 Proliferation Assay (CCK-8)

GC cells were seeded in 96-well plates at a density of 3000 (HGC-27) or 5000 (MKN-28) cells per well and treated with 10 μL of CCK-8 solution (Meilun Biotechnology Co., Ltd., Dalian, China) after being cultured for 22.5 h, 46.5 h, and 70.5 h. The absorbance of the cells was measured at 450 nm using a microplate reader according to the manufacturer’s instructions (Synergy4; BioTek, Winooski, VT, USA).

### 2.10. Transwell Assays

Transwell assays were performed using 24-well Transwells (8 μm pore size; Jet Bio-Filtration Co., Guangzhou, China) precoated with Matrigel (Corning, New York, NY, USA). The cells were diluted with serum-free medium so that the total number of cells in the upper chamber was 5–10 × 10^4^. The upper medium containing treatment reagent and cells but no serum. The bottom chamber was 600 μL medium containing 10% FBS. The cells were cultured in a constant 5% CO_2_ and 37 °C incubator for about 24 h, then fixed in methanol for 15 min, and stained with crystal violet for 15 min (Solarbio, Beijing, China). After drying in the chamber, cell invasion in different groups was recorded under a Zeiss microscope.

### 2.11. Wound Healing Assay

Cells were cultured in serum-containing medium in six-well plates. When the cells covered the bottom of the six-well plates, 3 or 4 scratches were made and serum-free medium containing siRNA and nc-siRNA was used to continue culture. After being incubated at 37 °C with 5% CO_2_ for 24 h, the images of the wounds were captured in 3 different areas of each group at each time period (0 h and 24 h), and the changes in the area of migration were calculated.

### 2.12. Microarray Analysis and ceRNA Network Analysis

CircRNA microarray was performed to investigate potential key circRNA in GC (Kang Chen Biotech, Shanghai, China) as we covered in our previous article [20].

TargetScan and miRanda were used to identify potential targets of microRNAs.

The GO (Gene Ontology) project was used to describe gene and gene product attributes in any organism, including biological process, cellular component, and molecular function domains. Meanwhile, we also performed pathway analysis, which involves mapping gene function to KEGG pathways. The lower the *p* value, the more significant the GO term; *p* ≤ 0.05 was our threshold.

### 2.13. Western Blotting (WB)

Strong RIPA lysis buffer (GenStar, Nanjing, China) was used to lyse cells, bicinchoninic acid (BCA) was used to quantify proteins (GenStar, Nanjing, China). Then, 10% SDS-PAGE and PVDF membrane (Millipore, Burlington, MA, USA) were performed to separate and transfer protein. Primary antibodies, including DAXX (Abways, Beijing, China), ERK1/2 (ABclonal, Wuhan, China), p-ERK (Cell Signaling Technology, Connors Farm, MA, USA), mTor (Cell Signaling Technology, USA), and actin (Cell Signaling Technology, USA) were incubated with membrane at 4 °C for more than 8 h. Then, secondary antibody (1:500) was incubated with membrane for 1 h at room temperature. Finally, the blots were visualized by Chemiluminescent HRP Substrate (Millipore, Burlington, MA, USA). ImageJ was used to analyze the results.

### 2.14. Dual Luciferase Reporter Assay

The wild-type (WT) and mutant (MUT) of hsa_circ_0044301 were designed by HanBio (Shanghai, China), and inserted into the pSI-Check2 vector. Then, this vector and hsa-miR-188-5p mimics or mimics negative control (nc) were co-transfected into 293T cells. Cells were collected and detected after transfection 48 h. Firefly Luciferase value was measured and recorded as an internal reference value, and Renilla Luciferase value was measured and recorded as reporter gene luminescence value.

### 2.15. RNA Immunoprecipitation (RIP) Assay

When the number of HGC-27 cells was sufficient, the cells were lysed with a lysate prepared according to manufacturer’s instructions and placed in a −80 °C refrigerator for use (Millipore, Billerica, MA, USA). The immunomagnetic beads were washed and 5 μg of target antibody (Argonaute2 (AGO2), DAXX, ERK1/2, and P-ERK antibodies) and negative control antibody IgG were added, respectively. Then, the RNA binding protein complex immunoprecipitation was performed. Finally, RNA was purified and the RNA in immunoprecipitation was analyzed by reverse transcription and qRT-PCR.

### 2.16. Immunofluorescence (IF) Assay

Treated cells were fixed with 4% polyformaldehyde, permeated with 0.5% Triton X-100, and blocked with 1% FBS. Then, the primary antibody was added at 1:50–200 and incubated overnight at 4 °C, followed by incubation with a 1:50–200 fluorescent secondary antibody for 30 min at 37 °C. DAPI was incubated for 5 min under dark conditions, and then pictures were taken under a Zeiss microscope.

### 2.17. Immunohistochemical Staining (IHC)

Immunodeficient experiments were performed by Servicebio (Wuhan, China). In brief, all tumor masses were fixed with 4% paraformaldehyde solution, paraffin embedded, xylene dewaxed, then rehydrated in graded ethanol, citric acid antigen repaired, endogenous peroxidase blocked, serum blocked, and antibody Ki67 (Servicebio, Beijing, China) incubated. Diaminobenzidine (DAB) was used for color rendering, hematoxylin was re-dyed, and finally the image was collected and analyzed under Zeiss microscope.

### 2.18. Mouse Tumorigenesis Model

The mouse model was constructed using HGC-27 cells, as described previously [21]. One week after injecting HGC-27 cells at density 1 × 10^7^ mL, total 200 μL of cell suspension, BALB/c nude mice were randomly divided into two groups according to tumor volume. One group was used for in vivo-specific siRNA-mediated circRNA-0044301 knockdown, and the other group was used for in vivo control studies. Three nude mice in each group were injected specific siRNA or control regent once every 2–3 days at a dosage of 5 nmol for four times in total. Tumor size of nude mice was measured and recorded before each injection and at the end of the experiment. The size of subcutaneous tumor was measured as follows: tumor volume (mm^3^) = (L × W^2^) × 0.5, where L represents the longest axis and W the shortest axis. About two weeks after treatment, nude mice were sacrificed and subjected to further investigation.

### 2.19. Statistical Analysis

Graphpad prism 6 and R software were used to analyze all data. When analyzing the data between the two groups, paired sample *t* test or independent sample *t* test was selected according to the data types between the two groups. The relationship between circRNA-0044301 (miRNA-188-5p) expression and patient survival was analyzed by Kaplan–Meier analysis and log-rank tests. Using Cox proportional risk regression for single-factor analysis, the risk ratio (HR) and 95% confidence interval (95% CI) were calculated. The data were analyzed and visualized using R software’s survival and survminer packages, as well as coxph commands. A *p* value less than 0.05 was considered statistically significant.

## 3. Results

### 3.1. CircRNA Microarray Results

To investigate the circRNA expression profile in GC tissues, we selected 3 pairs of GC (3 cancer tissues and 3 adjacent normal tissues) for analysis using circRNA microarray. The quality report is summarized in Appendix A. A total of 11,514 circRNA were identified, including 5775 upregulated and 5739 downregulated ones (Appendix A). More than 80% of them were from exonic. The proportions of intronic and sense-overlapping circRNA were nearly equal, consistent with the findings of recent studies (Appendix A) [22].

Twenty-five significantly differentially expressed circRNA(SDE-circRNA) were identified based on the following criteria: |fold change (FC)|≥ 2.0 and *p* < 0.05 (Figure 1A). Similarly, most of these SDE-circRNA were from exonic. We further analyzed the distribution of the SDE-circRNA on chromosomes, finding that some circRNA were significantly upregulated or downregulated in one chromosome (CHR17, CHR6, Chr19, and CHR12). The number of SDE-circRNA distributed on each chromosome was also different. For example, 8 SDE-circRNA were all derived from chromosome 17, with nearly half of the genes with upregulated expression (46%, 6 out of 13) distributed on this chromosome (Figure 1B). Given that chromosome 17 is associated with many cancers [23,24,25], therefore, it is also worth exploring the circRNA related to cancer from chromosome 17.

We then randomly selected five circRNA (hsa_circRNA_0000081 (downregulated), hsa_circRNA_0004662 (upregulated), hsa_circRNA_0008586 (upregulated), hsa_circRNA_0044301 (upregulated), and hsa_circRNA_404289 (upregulated)) from the 25 SDE-circRNA for preliminary studies according to FC (larger is better) and *p* (smaller is better) and validated them in 61 paired tissue samples. Results showed that the expression trends of hsa_circRNA_0004662, hsa_circRNA_0008586, and hsa_circRNA_0044301 in the tissue samples was consistent with the sequencing results (Figure 1C–G).

### 3.2. SDE-circRNA Were Shared between Our Results and GSE89143

Eight datasets were obtained with the key words “gastric cancer” and “circRNA” or “circRNA” (GSE122796, GSE116675, GSE83521, GSE100170, GSE93541, GSE89143, GSE77661, and GSE78092). The detailed information of these 8 datasets is available in Appendix A. The data of GSE122796 and GSE100170 could not be accessed; those of GSE116675 include GC and EV virus; those of GSE83521 are related to GC stage III; those of GSE93541 are related to plasma; and those of GSE77661 are related to many cancers. Only the data of GSE89143 and GSE78092, which are related to GC, could be obtained, and thus both of them were selected for comparative studies. Ultimately, we selected GSE89143 to compare our circRNA microarray results because it was published more recently (Appendix A). Using GSE89143, we identified 2558 circRNA between GC and normal tissues (Appendix A).

Using a Venn diagram (Figure 2), we found 2434 circRNA shared between our dataset and GSE89143 (Appendix A). Additionally, there were 4 circRNA with different expression patterns: hsa_circRNA_102039, hsa_circRNA_102110, hsa_circRNA_101947, and hsa_circRNA_101001; their IDs in circBase are hsa_circ_0043138, hsa_circ_0044301, hsa_circ_0041481, and hsa_circ_0025135, respectively. The details of the four circRNA are provided in Appendix A.

### 3.3. CeRNA Network, GO, and KEGG Analyses of Shared circRNA

Next, we performed functional prediction based on TargetScan and miRanda using the four shared SDE-circRNA. Results indicated that each circRNA targets many miRNA (Figure 3A). 

We next performed GO and KEGG analyses in http://www.geneontology.org, accessed on 31 May 2019. The former covers three domains: biological process (Figure 3B), cellular component (Figure 3C), and molecular function (Figure 3D). The top five biological processes were as follows: GO: 0071222 cellular response to lipopolysaccharide, GO: 0050829 defense response to Gram-negative bacteria, GO: 0071219 cellular response to molecules of bacterial origin, GO: 0032496 response to lipopolysaccharide, and GO: 0071216 cellular response to the biotic stimulus. The top five cellular components were GO: 0072559 NLRP3 inflammasome complex, GO: 0061702 inflammasome complex, GO: 0005759 mitochondrial matrix, GO: 1902494 catalytic complex, and GO: 0044445 cytosolic part. The top five molecular functions were GO: 0004303 estradiol 17-beta-dehydrogenase activity; GO: 0097153 cysteine-type endopeptidase activity involved in the apoptotic process; GO: 0033764 steroid dehydrogenase activity, acting on the CHOH group of donors, NAD or NADP as acceptor; GO: 0016229 steroid dehydrogenase activity; and GO: 0000166 nucleotide binding.

KEGG results showed the top 10 enrichment scores (−log_10_ (*p* value)) of the significantly enriched pathways (Appendix A). We found that insulin resistance was the most significant enrichment pathway and was related to the mTOR, TNF, and MAPK pathways (Appendix A). We should note that mTOR, TNF, and MAPK are all key molecular pathways in GC [26,27,28], especially the mTOR and MAPK pathways.

### 3.4. Correlation between circRNA-0044301 and Clinicopathological Indicators in Patients

Because hsa_circ_0044301 (circRNA-0044301) exhibited the lowest *p* value (*p* = 0.000587155) among the four shared SDE-circRNA and has been validated in 61 pairs of tissues (Figure 1F), we selected circRNA-0044301 for further investigation.

Further analysis showed that circRNA-0044301 expression was not significantly associated with patients’ clinicopathological parameters (Table 1 and Appendix A). To understand the association between circRNA-0044301 expression and patient survival, we followed the enrolled patients until February 2022, obtaining data of up to 45 patients with valid survival time and survival conditions (Appendix A). Grouping according to the median circRNA-0044301 expression demonstrated that patients in the low circRNA-0044301 expression group had longer survival and higher five-year survival rates (Figure 4). Thus, we performed univariate Cox regression analysis using age, gender, stage, tumor size, and circRNA-0044301 expression as variables. From Table 2, circRNA-0044301 expression was significantly associated with patients’ survival, while it could not be an independent factor for the results of multivariate Cox regression analysis (Table 2).

However, due to the significant differential expression of circRNA-0044301 in tissues and its relationship with patient prognosis, we decided further explore whether circRNA-0044301 can influence the function of GC cells.

### 3.5. CircRNA-0044301 Influences the Function of GC Cells In Vitro and In Vivo

Firstly, we investigated circRNA-0044301 expression in 6 cell lines: GES-1, HGC-27, AGS, MKN-45, MKN-28 (MKN-74), and MGC-803 (Appendix A), which was consistent with the microarray results. As illustrated in Appendix A, circRNA-0044301 expression differed among five cancer cell lines. Interestingly, we found that HGC-27 was a non-differentiated cell line, MGC-803 and MKN-45 exhibited low differentiation, MKN-28 was moderately differentiated, and AGS exhibited medium to low differentiation, as described at https://web.expasy.org/cellosaurus, accessed on 13 April 2021.

Therefore, we selected HGC-27 and MKN-28 for further investigation. FISH experiments indicated that circRNA-0044301 was present in both the cytoplasm and nucleus and was mainly located in the cytoplasm (Figure 5A). RNase R (Figure 5B) and actinomycin D assays (Figure 5C) showed that circRNA-0044301′s stability and half-life were significantly higher than those of linear RNAs. Sanger sequencing (Appendix A) results demonstrated that our qRT-PCR product was accuracy, indicating that our designed primers could be used for further investigation.

Next, to investigate the function of circRNA-0044301 in GC, we performed siRNA assays in HGC-27 and MKN-28 cells. The effect of siRNA on circRNA-0044301 was determined by qRT-PCR (Appendix A). According to CCK-8 (Figure 5D), wound healing (Figure 5E), and Transwell (Figure 5F) assays, interference of circRNA-00044301 decreased the proliferation, migration, and invasion of GC cells.

We further established a subcutaneous tumorigenesis model to further validate the role of circRNA-00044301 in vivo. SiRNA-vivo was used to suppress circRNA-00044301 expression (Figure 5G). Results showed that tumor volume was significantly reduced in the nude mice after siRNA injection compared with that in the control group (Figure 5H). The expression of tumor proliferation-related protein Ki67 in the circRNA-00044301 knockdown group was also significantly lower than that in the control group (Figure 5I). These results indicated that circRNA-0044301 could be a potential treatment target.

### 3.6. MiRNA-188-5p Was Sponged by circRNA-0044301 and Was Associated with GC Patients’ Prognosis

We performed a RIP assay with an anti-AGO2 antibody and IgG in HGC-27 cells to investigate whether circRNA-0044301 could be combined with them. As expected, the percentage of circRNA-0044301 pulled down by anti-AGO2 was significantly higher than that in the anti-IgG group (Figure 6A), suggesting that circRNA-0044301 could be a ceRNA participating in GC development.

Combining the above results, hsa-miR-488-5p, hsa-miR-623, hsa-miR-670-3p, hsa-miR-520f-3p, hsa-miR-127-5p, hsa-miR-128-1-5p, hsa-miR-128-2-5p, hsa-miR-188-5p, and hsa-miR-146a-3p were predicted to be targets of circRNA-0044301 (Appendix A). Therefore, we first selected five miRNA (hsa-miR-623, hsa-miR-520f-3p, hsa-miR-127-5p, hsa-miR-188-5p, and hsa-miR-146a-3p) with relatively more reports than the other miRNAs to detect their expression in the si-circRNA-0044301 and nc-circRNA-0044301 groups. All of them were upregulated when circRNA-0044301 was knocked down (Figure 6B); however, their expression in the cells varies. Among them, the expression of hsa-miR-188-5p (miRNA-188-5p) in each GC cell was significantly downregulated compared with normal epithelial cells (GSE-1) (Appendix A). In addition, previous study reported that miRNA-188-5p was related to many diseases [29,30,31,32]. Therefore, we selected miRNA-188-5p for further investigation.

Firstly, we used a double luciferase reporter genetic assay to determine the relationship between miRNA-188-5p and circRNA-0044301. We found that in the circRNA-004430-mu strain, the use of mimics-miRNA-188-5p and nc-miRNA-188-5p did not affect the luciferase fluorescence activity of circRNA-0044301. In contrast, in the circRNA-004430-wt strain, the fluorescence activity of circRNA-0044301 was significantly reduced when using mimics-miRNA-188-5p compared with that of nc-miRNA-188-5p, indicating a correlation between circRNA-0044301 and miRNA-188-5p (Figure 6C).

Then, we acquired the TCGA-STAD miRSeq dataset from gdac.broadinstitute.org, (Appendix A). Result showed that the expression of miRNA-188 was upregulated in 436 GC tissues compared with 41 normal tissues (*p* < 0.0001, there is no miRNA-188-5p in these sequencing data, which is a common variant of miRNA-188) (Appendix A).

To perform a comparative analysis to the disease stage, we classified stages based on the 7th edition (The H column of Appendix A) [33]. From Table 3, we could find that the high and low expression group of miRNA-188 was statistically significant in the longest tumor dimension (according to the median expression level of miRNA-188 divided into high and low groups. Because the longest tumor diameter in the TCGA dataset is 3.5 (Appendix A), 1.5 is used as the cut-off value for the longest tumor diameter in GC patients in TCGA).

Next, survival information of patients was obtained at https://xenabrowser.net/datapages/, accessed on 31 May 2019 (Appendix A), and interesting results were found in age groups: in the group less than or equal to 56 years of age, the higher the expression of miRNA-188, the longer the survival (Figure 6D); in the group greater than 75 years of age, the lower the expression of miRNA-188, the longer the survival (Figure 6F). The expression of miRNA-188 in the middle age group was not related to survival (Figure 6E), indicating that the expression of miRNA is dynamic and may be affected by host’s own conditions, such as age, tumor microenvironment, and other factors.

These results suggest that circRNA-0044301 could serve as a molecular sponge for miRNA-188-5p, and miRNA-188-5p (miRNA-188) was correlated with the progression of GC.

### 3.7. CircRNA-0044301 Participated in the Progression of GC via Regulating the miRNA-188-5p/DAXX Axis

As shown in Appendix A, both DAXX and DHRS12 were the targets of miRNA-188-5p. Therefore, we investigated their expression in mimic-miRNA-188-5p and si-circRNA-0044301-treated cells. Results showed that DAXX expression was reduced in the mimic-miRNA-188-5p and si-circRNA-0044301 groups compared to that in the control groups, while the expression level of DHRS12 was not affected by miRNA-188-5p mimics (Figure 7A). In a further experiment combining mimic-miRNA-188-5p and si-circRNA-0044301, it was found that the expression level of DAXX further reduced, DHRS12’s did not change significantly (Appendix A), and RIP-qRT-PCR experiment results indicated that DAXX could directly interacted with miRNA-188-5p (Figure 7B). Since the expression of DAXX was downregulated in mimic-miRNA-188-5p- and si-circRNA-0044301-treated cells, directly interacted with miRNA-188-5p, and significantly affected by circRNA-0044301, we selected DAXX for further investigation.

We first investigated the association between DAXX and circRNA-0044301 in 12 pairs of tissues, finding that their expression was highly related (Figure 7C; R^2^ = 0.9218, *p* < 0.0001). Moreover, we found that DAXX protein could also combine with circRNA-0044301 directly (Figure 7D). In addition, DAXX expression was upregulated in GC cells compared with that in normal cells, consistent with the expression of circRNA-0044301 (Appendix A). Although DAXX was affected by miRNA-188-5p, it showed a direct correlation with circRNA-0044301 at both the RNA and protein levels. Therefore, we wondered whether DAXX is also functionally affected by circRNA-0044301.

After constructing the DAXX siRNA cell model successfully (Appendix A), we performed CCK-8 (Figure 7E), wound healing (Figure 7F), and Transwell (Figure 7G) assays with nc-DAXX and si-DAXX transfected into HGC-27 cells, which indicated that DAXX promoted the proliferation, migration, and invasion of GC and si-DAXX could inhibit this effect. We further explored the effects of si-circRNA-0044301 in association with si-DAXX on the proliferation and invasion of GC cells and found that the combination of the two more obviously inhibited cell proliferation and invasion (Figure 7H,I), which suggested that DAXX was also functionally affected by circRNA-0044301. In other words, circRNA-0044301 influences GC process is likely by regulating miRNA-188-5p/DAXX axis.

### 3.8. Knockdown of circRNA-0044301 Inhibited ERK1/2 Expression and Enhanced the Sensitivity of Cells to 5-FU

As mentioned above, circRNA-0044301 may be related to the mTOR and MAPK pathways, and mTOR, p-ERK, and ERK1/2 are the key proteins in the mTOR and MAPK families; hence, we investigated the expression of mTOR, p-ERK, and ERK1/2 in MKN-28 and HGC-27 cells with si-circRNA-0044301 and nc-circRNA-0044301 using WB (Appendix A). We found that the expression of the mTOR protein was not affected by si-circRNA-0044301 but ERK1/2 and p-ERK, so we investigated whether ERK1/2 and p-ERK could pull down circRNA-0044301 through RIP experiments. Results indicated that both of them could significantly pull down circRNA-0044301 (Figure 8A). To further explore the relationship among ERK1/2, p-ERK, and circRNA-0044301, we investigated the expression of ERK1/2, p-ERK, and circRNA-0044301 in four pairs of GC tissue samples, namely #1, #2, #3, and #4, finding that the expression of ERK1/2 and p-ERK proteins was downregulated in all patients. At the same time, the expression of circRNA-0044301 was downregulated in #1, #2, and #4, but upregulated in #3 (Figure 8B), suggesting that their expression trend in tissues was slightly correlated.

Given the above association between ERK1/2 (p-ERK) and circRNA-0044301, we hypothesized that silencing of circRNA-0044301 would impact the inhibitory effect of the ERK1/2 inhibitor. Therefore, we treated the cells with the ERK1/2 inhibitor GDC-0994 and after confirming the IC50 of GDC-0994 in the cells (Appendix A), we clarified the relationship between GDC-0994 and si-circRNA-0044301 through IF (Figure 8C) and WB (Appendix A). The results showed that si-circRNA-0044301 could enhance the effect of the inhibitor on ERK1/2.

Because ERK1/2 and p-ERK are key proteins in the MAPK signaling pathway and the investigation of MAPK in this study is based on the prediction of the ceRNA network that the pathway in which the downstream target gene of the miRNA may be involved includes MAPK. Therefore, we further investigated whether DAXX is related to ERK1/2. Results indicated that DAXX not only has an effect on the transcription level of ERK1/2 (Appendix A), but also has an impact on the translation level of ERK1/2(Figure 8D); and our RIP experiments showed that there was a direct interaction between DAXX and ERK1/2 (Figure 8E). Thus, we inferred that one potential mechanism of circRNA-0044301 on ERK1/2 is likely to be through DAXX.

Finally, we explored the role of circRNA-0044301 on 5-FU. After determining the IC_50_ of the commonly used clinical chemotherapy drug 5-FU (Appendix A), proliferation experiment suggested that si-circRNA-0044301 could enhance the sensitivity of cells to 5-FU (Figure 8F), which again illustrates the potential of circRNA-0044301 to be a therapeutic target for GC.

## 4. Discussion

In this study, circRNA-0044301 was investigated, which is from exonic on chr17 and whose expression is upregulated in GC. According to our clinical data analysis, the patients with high expression of circRNA-0044301 had a poor prognosis; moreover, when we treated tumors with circRNA-0044301-specific siRNA in the nude mouse model constructed for GC, the tumor volume was significantly reduced compared with that in the control group. These results demonstrated that circRNA-0044301 has the potential to be a target for GC treatment.

In terms of biogenesis, circRNA can mainly be classified into three categories: exon-exon circRNA (ecircRNA) [34], intron-intron circRNA (icircRNA) [35], and exon-intron circRNA (EIcircRNA) [36]. There are several theories about the biogenesis of these three types: intron pairing [9], snRNAPs (small nuclear ribonucleoproteins) [37], and RBPs [38] mainly form ecircRNA and eicirRNAs; the lasso structure can form all types [35]. EcircRNA are mainly found in the cytoplasm [39], and the other two are mainly found in the nucleus [40]. The different locations result in different functional mechanisms. CircRNA-0044301 is mainly present in the cytoplasm. AGO2-RIP experiments have also shown that circRNA-0044301 could act as a ceRNA. In addition, the dual luciferase assay further indicated that miRNA-188-5p had a direct relationship with circRNA-0044301, which confirmed our speculation.

Some reports have indicated that miRNA-188-5p contributes to choroidal neovascularization [29] and that downregulation of miRNA-188-5p contributes to the pathogenesis of Alzheimer’s disease [30] and affects the development of hepatocellular carcinoma [31,32]. It has also been reported that miRNA-188-5p was upregulated in retinoblastoma and can promote EMT by targeting ID4 through the Wnt/β-catenin signaling pathway [41]. These results indicate that miRNA-188-5p expression differs in different diseases. In our results, miRNA-188-5p expression was downregulated in GC cells, while miRNA-188 was upregulated in GC tissues compared with normal tissues according to TCGA database and its expression level dynamically changes with the age of the patients. The downstream target gene of miRNA-188-5p, DAXX, a death domain-associated protein, is a multifunctional protein that has been implicated in pro-apoptosis, anti-apoptosis, and transcriptional regulation processes [42]; acts as a tumor suppressor in osteosarcoma by mediating the deposition of histone H3.3 [43]; and is involved in pancreatic neuroendocrine tumors [44] and glioblastoma multiforme [45], as its mutation corresponds to the presence of alternative lengthening of telomeres. In addition, DAXX is functional in ovarian cancer by activating the ERK signaling pathway [46] and in breast cancer by impeding DNA damage repair [42]. However, it is unclear whether DAXX is associated with GC and what the underlying molecular mechanism may be. In our study, we found that DAXX is upregulated in GC; affects the proliferation, invasion, and migration of GC cells; and is associated with circRNA-0044301. To our knowledge, this is the first study of circRNA-0044301 affecting GC progression via miRNA-188-5p/DAXX axis.

Although the role and mechanism of circRNA-0044301 as ceRNA in GC have been further explored, the function of circRNA is not limited to regulating downstream targets by acting as miRNA molecular sponges but they can also act as a regulator of the transcription or expression of parent genes. For example, a study found that icircRNA could regulate the transcriptional activity of RNA polymerase II through RNA-RNA interactions in the nucleus, facilitating the transcription of parent genes [35]. In addition, circRNA can also interact with proteins, such as the circFoxo3-CDK2-p21 binary complex. circFoxo3 can enhance p21 inhibition of CDK2 and cell cycle protein A (cyclin A) and cyclin E binding, resulting in cell cycle stagnation or apoptosis [47]. In our study, we found that circRNA-0044301 could be pulled down by the key proteins ERK1/2 and p-ERK in the MAPK signaling pathway and could affect the inhibitory effect of GDC-0994 on ERK1/2. GDC-0994, an oral inhibitor of ERK1/2, is widely used in various experimental studies, especially in combination with other inhibitors to explore its ability to treat diseases [48,49]. Fortunately, in our preliminary study, we found that si-circRNA-0044301 combined with GDC-0994 could further inhibit cell activity; moreover, knockdown circRNA-0044301 could also enhance the sensitivity of cells to 5-FU, the first-line chemotherapy drug [50]. These results suggested that circRNA-0044301 has some therapeutic potential both alone and in combination.

In addition to the above function, circRNA could also encode proteins. The internal ribosome entry site (IRES) is the structure necessary for a molecule to not rely on the 5’ end cap to translate the protein independently [51]. Studies have shown that some circRNA contain IRES sites, such as circ-FBXW7 [52]. Some experiments have shown that there are also rich m6A modifications in circRNA which are related to coding proteins but also affect the body’s innate immunity [53]. Additionally, circRNA does not exist only in tissues and plasma, as numerous circRNA have also been recently found in exosomes [54,55]. This not only enriches our understanding of the function of circRNA but also stimulates the interest in studying circRNA. However, there are still many unknowns about the carcinogenic mechanism of circRNA. Researchers have reported that the structure of circRNA and the binding proteins are dynamically changing [56], which further complicate the mechanism of circRNA and their associated axes function in disease. Therefore, the field still needs further research and insight.

At present, in addition to research on small RNAs as a disease treatment target [57], some researchers have been working on the targeted transport of circRNA, such as the use of extracellular vesicles and nanomaterials to transport circRNA to specific sites, to explore the role of circRNA in specific locations [58,59], which extends the direction of circRNA drug conversion, and is also the direction of our efforts to improve the diagnosis and treatment effect of GC.

Admittedly, this study also has many limitations. Firstly, the sample size was small, which directly affects our inference of the association between circRNA-0044301 and basic patient information and clinical indicators. Although we confirmed the expression of circRNA-0044301 using 61 paired tissues, in the analysis of clinical data of patients, we grouped high and low expression levels with the median expression level of circRNA-0044301, which resulted in a small number of samples in each group. Therefore, more samples were needed to verify our conclusions. Secondly, the animal model is not the ideal. We should construct an orthotopic transplanted tumor model, because the growth environment of this model is more similar to that of clinical patients [60]. However, only the subcutaneous tumor model was constructed in this study. Next, we did not perform additional assays to investigate the mechanism underlying circRNA-0044301 function in GC. The ceRNA mechanism involves more than one circRNA and one miRNA, but we only selected miRNA-188-5p for further study. Additionally, how and to what extent they interact, such as how circRNA-0044301 and miRNA may be unilateral or bidirectional, and the relationship between circRNA-0044301 and other key proteins of the related pathway or phosphorylated proteins of the mTOR pathway also requires further investigation. Moreover, the etiology of GC is complex, involving a multifactorial process resulting from the combination of genetic susceptibility and environmental risk factors [61]. *Helicobacter pylori* infection is a risk factor for GC and has been classified as a level Ι carcinogen by the World Health Organization [62]. Research reports that toxic factors of *H. pylori* play an important role in this cancerous process, where cytoxin-associated gene A (CagA) expression is a possible cause of the different clinical outcomes after infection [63]. Although we infected cells with CagA protein and found that circRNA-0044301 expression could be affected (Appendix A), we did not go any further into how they work.

## 5. Conclusions

We found that the expression of circRNA-0044301 was upregulated in GC. It was observed to influence GC progression both in vitro and in vivo, and was inversely correlated with patient prognosis. In addition, circRNA-0044301 could regulate the role of the downstream target DAXX by acting as a molecular sponge of miRNA-188-5p in GC. It also affected pathway inhibitors and had an impact on the effect of action of the chemotherapy drug 5-FU, as well as be influenced by the main toxic factor CagA of *H. pylori*. These results indicate that circRNA-0044301 has potential as a therapeutic target in GC.

## Figures and Tables

**Figure 1 cancers-14-04183-f001:**
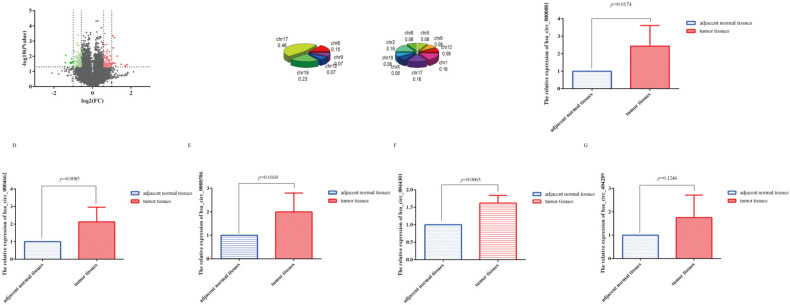
The results of our circRNA microarray. (**A**) Volcano plot of differentially expressed circRNAs showed that the expression of 5775 and 5739 circRNAs was upregulated and downregulated, respectively; pale black indicates no differential expression; dark red indicates significantly upregulated expression, and dark green indicates downregulated expression with absolute FC (fold change) greater than 2 and *p* value less than 0.05; pale red shows upregulated expression, and pale green shows downregulated expression with absolute FC greater than 1.5 and *p* value less than 0.05. (**B**) Dysregulated circRNAs located on different chromosomes; the left 3D pie plot shows the location of significantly upregulated expression of 13 circRNAs, and the right plot shows the location of significantly downregulated expression of 12 circRNAs. (**C**–**G**) QRT-PCR of the relative expression of hsa_circRNA_0000081, hsa_circRNA_0004662, hsa_circRNA_0008586, hsa_circRNA_0044301, and hsa_circRNA_404289 in 61 pairs of gastric cancer tissues, respectively (paired sample *t* test was performed using the 2^−ΔΔCT^ value of each pair of samples).

**Figure 2 cancers-14-04183-f002:**
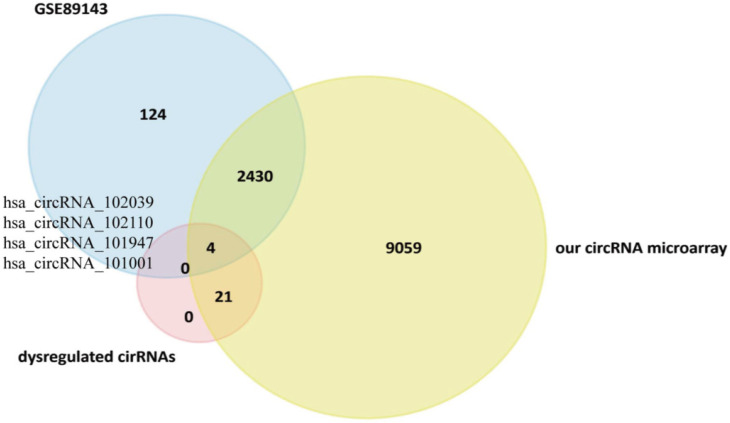
Schematic illustration of the overlapping circRNAs in GC, as identified by GSE89143, our circRNA microarray, and our significantly dysregulated circRNAs.

**Figure 3 cancers-14-04183-f003:**
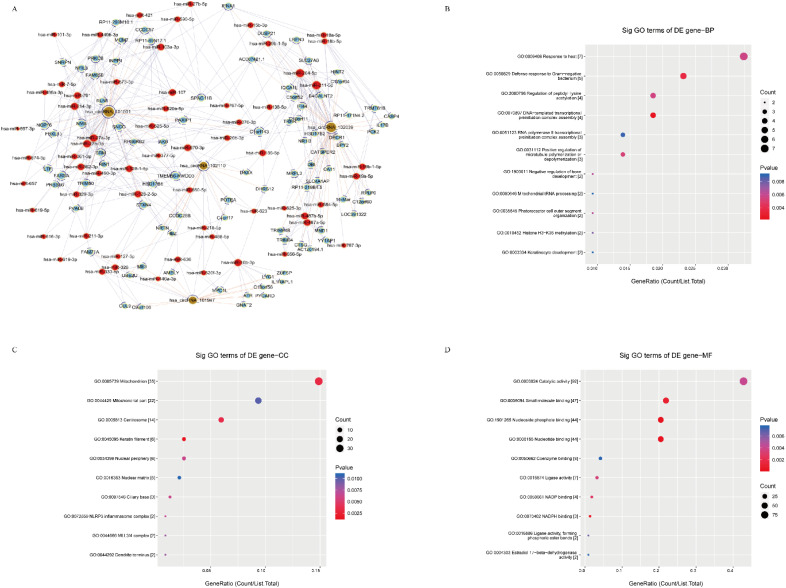
CeRNA network and GO analyses of shared circRNAs. (**A**) CeRNA network among hsa_circ_0043138, circRNA-0044301, hsa_circ_0041481, and hsa_circ_0025135 and their targeted miRNAs and genes (nodes: red nodes are microRNAs; light-blue nodes are protein-coding RNAs; light-green nodes are noncoding RNAs; brown nodes are circular RNAs; edges: edges with a T-shaped arrow represent direct relationships; edges without an arrow represent indirect relationships (ceRNA relationship)). (**B**) Biological processes of the four circRNAs. (**C**) Cellular components of the four circRNAs. (**D**) Molecular functions of the four circRNAs (“GO. ID” stands for the ID of the GO term; “Count” stands for the number of DE genes associated with the listed GOID; “*p* value” stands for the significance testing value of the GOID; “Enrichment Score” stands for the enrichment score value of the GOID, which equals (−log_10_(*p* value)).

**Figure 4 cancers-14-04183-f004:**
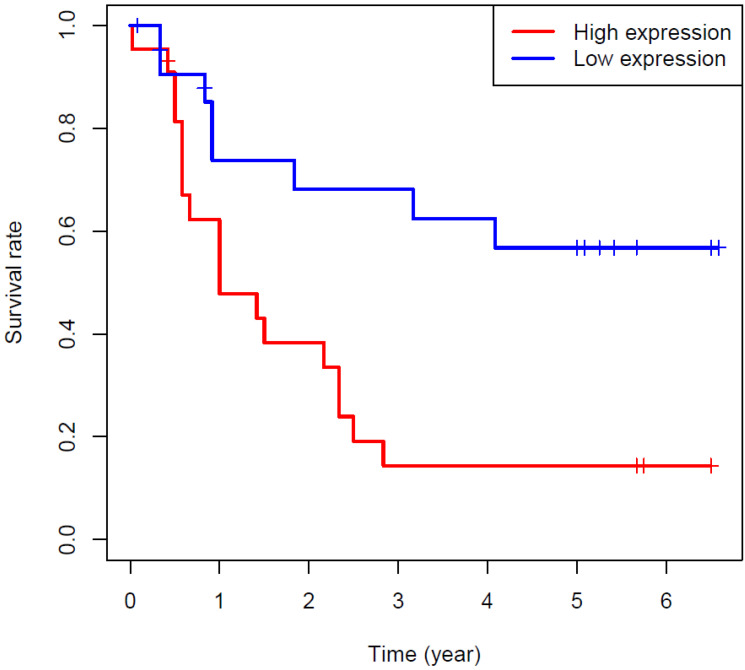
Survival curve of GC patients in the high- and low circRNA-0044301 expression groups showed that high circRNA-0044301 expression level was significantly associated with a poor prognosis of GC patients (*p* = 0.006).

**Figure 5 cancers-14-04183-f005:**
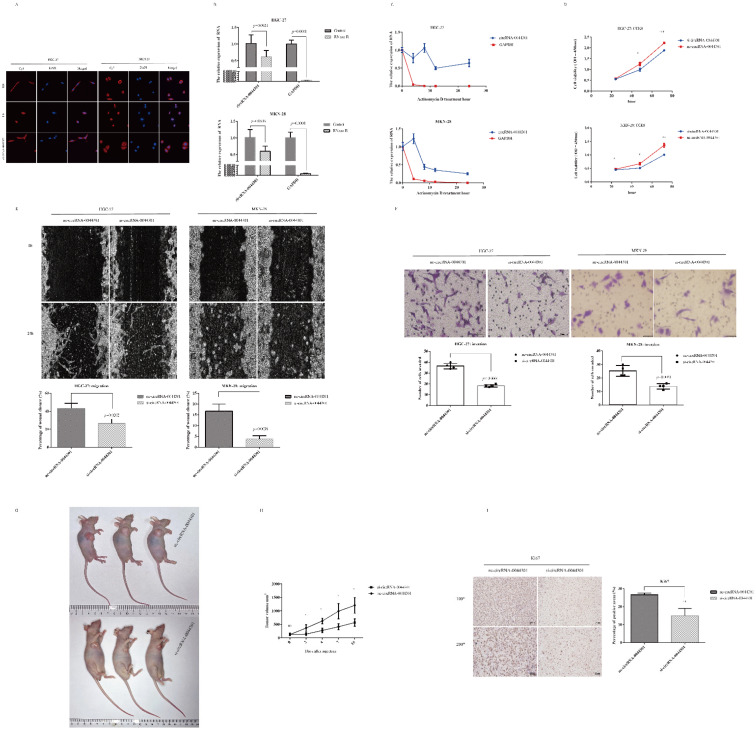
The characteristics and function of circRNA-0044301 on cells in vitro and in vivo. (**A**) FISH assay to detect the subcellular localization of circRNA-0044301 in HGC-27 and MKN-28 (The samples were imaged at 60× magnification). (**B**) RNase R treatment to determine the circular nature of circRNA-0044301 in GC cell lines. (**C**) Actinomycin D assay to explore the half-life of circRNA-0044301 and linear RNA in HGC-27 and MKN-28. (**D**) CCK-8 assay for the viability of HGC-27 cells (**top**) and MKN-28 cells (**bottom**) in the si-circRNA-0044301 group and nc-circRNA-0044301 group (***: *p* < 0.001; **: *p* < 0.01; *: *p* < 0.05; ns: no difference). (**E**) Wound healing assay for the cell migration ability of GC cells after circRNA-0044301 knockdown over time (0 h and 24 h), and the percentage of wound closure was statistically analyzed (The samples were imaged at 100× magnification. Scale bar = 100 μm). (**F**) Results of the Transwell assay of HGC-27 cells (**top**) and MKN-28 cells (**bottom**) transfected with si-circRNA-0044301 and nc-circRNA-0044301 (the samples were imaged at 200× magnification. Scale bar = 50 μm). (**G**) Subcutaneous tumorigenesis model, specific siRNA transfection was adopted to reduce circRNA-0044301 levels in tumors. (**H**) The size of tumor volume was measured in vivo every 2–3 days using Vernier caliper (*: *p* < 0.05; ns: no difference); (**I**) IHC assay demonstrated the level of Ki67 in control siRNA and specific siRNA-treated groups (**: *p* < 0.01).

**Figure 6 cancers-14-04183-f006:**
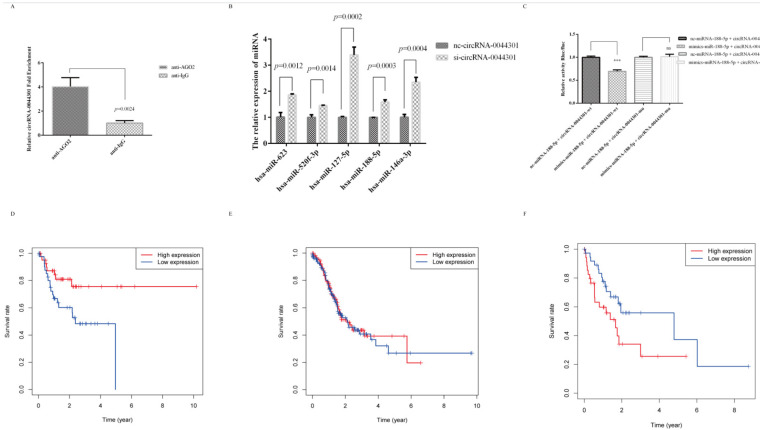
MiRNA-188-5p was sponged by circRNA-0044301 and was associated with GC patients’ prognosis. (**A**) AGO2-RIP assay results of circRNA-0044301 in HGC-27. (**B**) qRT-PCR of the expression of miRNAs in si-circRNA-0044301 and nc-circRNA-0044301 groups. (**C**) Luciferase reporter assay experiment in 293T cells co-transfected with circRNA-0044301 (mu or wt) and miRNA-188-5p (mimics or nc); the luciferase activity of the 293T cells transfected with circRNA-0044301 wt (wt-circRNA-0044301) sequence was significantly reduced but no significant inhibitory effect was observed in those transfected with the circRNA-0044301 mu sequence (mu-circRNA-0044301), (***: *p* < 0.001; ns: no difference). (**D**) Survival curve of GC patients less than or equal to 56 years in the high- and low miRNA-188 expression groups showed that a low miRNA-188 expression level was significantly associated with a poor prognosis (*p* = 0.017). (**E**) Survival curve of GC patients more than 56 and less than or equal to 75 years in the high and low miRNA-188 expression groups showed no difference (*p* = 0.859). (**F**) Survival curve of GC patients more than 75 years old in the high and low miRNA-188 expression groups showed that a high miRNA-188 expression level was significantly associated with a poor prognosis of GC patients (*p* = 0.048).

**Figure 7 cancers-14-04183-f007:**
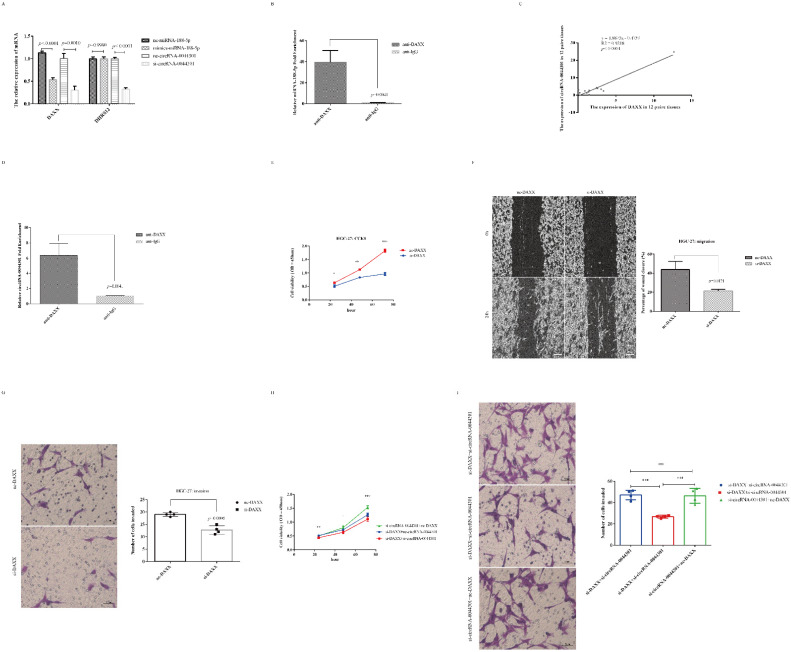
The function of DAXX and the association between DAXX and circRNA-0044301. (**A**) qRT-PCR analysis of DAXX and DHRS12 expression in mimics-miRNA-188-5p or nc-miRNA-188-5p and si-circRNA-0044301 or nc-circRNA-0044301 transfected cells. (**B**) RIP-qRT-PCR experiments confirmed the interaction between miRNA-188-5p and DAXX. (**C**) Correlation analysis of DAXX expression level and circRNA-0044301 expression level; their expression was positively correlated (*p* < 0.0001). (**D**) RIP-qRT-PCR experiments confirmed the interaction between circRNA-0044301 and DAXX. (**E**) CCK-8 assay of the cell viability of HGC-27 cells with si-DAXX or nc-DAXX transfection (***: *p* < 0.001; **: *p* < 0.01; *: *p* < 0.05). (**F**) Wound healing assay for the cell migration ability of GC cells after knockdown of DAXX over time (0 h and 24 h), and the percentage of wound closure was statistically analyzed (the samples were imaged at 50× magnification. Scale bar = 200 μm). (**G**) Results of Transwell assays with si-DAXX and nc-DAXX transfection into HGC-27 cells (the samples were imaged at 200x magnification. Scale bar = 50 μm). (**H**) CCK-8 assay for the cell viability of GC cells after si-DAXX, si-circRNA-0044301, and co-transfection into HGC-27 cells; the viability of the GC cell lines in the co-transfection group was significantly lower than that in the single transfection group (***: *p* < 0.001; **: *p* < 0.01; *: *p* < 0.05). (**I**) Transwell assay for the cell invasion ability of GC cells after si-DAXX, si-circRNA-0044301, or co-transfection into HGC-27 cells; the invasion ability of the GC cell lines in the co-transfection group was significantly lower than that in the single transfection group (the samples were imaged at 200x magnification. Scale bar = 50 μm. ***: *p* < 0.001).

**Figure 8 cancers-14-04183-f008:**
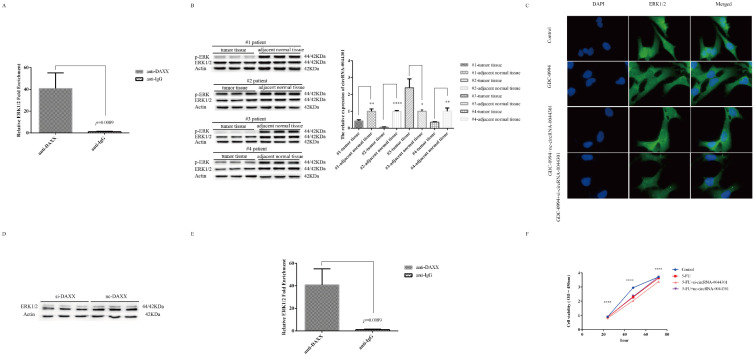
Knockdown of circRNA-0044301 inhibited ERK1/2 expression and enhanced the sensitivity of cells to 5-FU. (**A**) RIP-qRT-PCR experiments confirmed the interaction between circRNA-0044301 and ERK1/2 or p-ERK. (**B**) The trends of ERK1/2 and p-ERK protein expression were generally consistent with the trends of circRNA-0044301 expression in four GC patients (only in patient #3 was the expression trend reversed, but it was consistent in the other three patients. ****: *p* < 0.0001; **: *p* < 0.01; *: *p* < 0.05). (**C**) IF results of ERK1/2 in cells after single GDC-0994 or combined transfection with si-circRNA-0044301 into cells; the ERK1/2 protein expression was inhibited after the addition of GDC-0994 compared with that in the control group, and when GDC-0994 was transfected with si-circRNA-0044301 into cells, ERK1/2 protein expression was further inhibited compared with that in the control group (The samples were imaged at 1000× magnification). (**D**) WB results of ERK1/2 with si-DAXX treated cells. (**E**) RIP-qRT-PCR experiments confirmed the interaction between ERK1/2 and DAXX. (**F**) CCK-8 assay for the cell viability of GC cells after 5-FU drug and combined treatment with si-circRNA-0044301 of HGC-27 cells (****: *p* < 0.0001). The whole western blots are shown in Appendix A.

**Table 1 cancers-14-04183-t001:** Analysis of the correlation between circRNA-0044301 expression and clinicopathological parameters (*χ*^2^ Test).

Characteristics	Variable	Number (61)	CircRNA-0044301 Expression		*p*
			Low	High	
Age (years)	≤65	22	14	8	0.0862
	>65	37	15	22	
	NA	2			
Gender	female	20	7	13	0.1194
	male	39	22	17	
	NA	2			
AJCC (Version Type 8th)	stage I + II	5 + 24	4 + 13	1 + 11	0.1892
	stage III + IV	26 + 3	11 + 1	15 + 2	
	NA	3			
Tumor size (cm)	≤4.5	29	17	12	0.1444
	>4.5	28	11	17	
	NA	4			
Family history	yes	3	2	1	0.551
	no	45	22	23	
	NA	13			
History of hypertension	yes	25	11	14	0.4129
	no	29	16	13	
	NA	7			
Blood type	A	18	11	7	0.689
	AB	4	2	2	
	B	10	4	6	
	O	22	10	12	
	NA	7			

**Table 2 cancers-14-04183-t002:** Independent prognosis analysis of clinicopathological parameters and circRNA-0044301.

Parameters	Univariate Cox Regression			Multivariate Cox Regression		
	HR	95% CI	*p*	HR	95% CI	*p*
Age	10.363	2.114–50.804	0.004	6.781	1.281–35.909	0.024
Gender	0.744	0.177–3.134	0.687	0.531	0.110–2.576	0.432
Stage	8.118	1.828–30.042	0.006	4.832	0.972–24.033	0.054
Tumor size (cm)	6.323	1.475–27.101	0.013	2.353	0.430–12.882	0.324
circRNA-0044301	2.301	1.331–3.978	0.003	1.640	0.855–3.144	0.136

**Table 3 cancers-14-04183-t003:** Analysis of the correlation between miRNA-188 expression and clinicopathological parameters (*χ*^2^ Test).

Characteristics	Variable	Number (436)	MiRNA-188 Expression		*p*
			Low	High	
Age (years)	≤65	194	99	95	0.6666
	>65	237	116	121	
	NA	5			
Gender	female	155	87	68	0.0573
	male	281	131	150	
AJCC (Version Type 7th)	I + II	57 + 163	22 + 77	35 + 86	0.1228
	III + IV	158 + 30	84 + 15	74 + 15	
	NA+ IIB or IIIB	27 + 1			
Longest dimension (cm)	≤1.5	111	47	64	0.0255
	>1.5	121	69	52	
	NA	204			
Family history	No	317	158	159	0.9896
	Yes	18	9	9	
	NA	101			
Race category	White	273	137	136	0.3484
	Asian	88	46	42	
	Black or African American	13	4	9	
	NA	62			

## Data Availability

The processed data that support the findings of this study are available from the corresponding author.

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
