# Peer review of "Hsa_circ_0044301 Regulates Gastric Cancer Cell’s Proliferation, Migration, and Invasion by Modulating the Hsa-miR-188-5p/DAXX Axis and MAPK Pathway"

_cancers, 2022, doi:10.3390/cancers14174183_

Round 1
Reviewer 1 Report
Gastric cancer is relatively common and this study is only based on 61 patient samples. This reviewer does not think that the sampling size is adequate.
The authors find that circRNA-0044301 was elevated in gastric sancer and high levels were associated with decreased survival. This circRNA-0044301 specifically excluding 5 other circRNAs because it appeared to have the best results. I suggest that the authors provide some prospective evidence in new patients with gastric cancer to see if tissue and blood levels of circRNA-0044301 can prospectively diagnose gastric cancer and determine prognosis. This will make the study much more important to the reader and patient care. Hopefully it will diagnose gastric cancer and determine prognosis. Inhibition of circRNA-0044301 inhibited gastric tumor growth in a mouse model but the tumor still grew, albeit at a decreased rate. I think that the subcutaneous tumor model is not ideal because obviously the stomach is within the abdomen. Is this subcutaneous tumor relevant?
Author Response
- Gastric cancer is relatively common and this study is only based on 61 patient samples. This reviewer does not think that the sampling size is adequate.
Answer: Yes, you’re right, gastric cancer is relatively common, and our research group is still continuing to collect samples. However, the ethical review is very strict, and sampling must be obtained after ethical approval and patient consent. So we can only collect gastric cancer tissues for a fixed period of time. In addition, only those who met our inclusion criteria could be studied in this study. The inclusion criteria were: Patients with stomach cancer must have had their first stomach cancer surgery when they were studied, and had not received any radiation or chemotherapy before that. This further reduces the number of cases of study subjects. So in the end, there were 61 pairs of stomach cancer patients who met the criteria. We also regret this
- The authors find that circRNA-0044301 was elevated in gastric sancer and high levels were associated with decreased survival. This circRNA-0044301 specifically excluding 5 other circRNAs because it appeared to have the best results. I suggest that the authors provide some prospective evidence in new patients with gastric cancer to see if tissue and blood levels of circRNA-0044301 can prospectively diagnose gastric cancer and determine prognosis. This will make the study much more important to the reader and patient care. Hopefully it will diagnose gastric cancer and determine prognosis.
Answer: Thank you for your wishes and suggestion, and our research group is currently in the process of sampling.
However, the hospital's ethical review is very strict, and samples must be collected after ethical approval and patient consent. Therefore, we can only collect gastric cancer tissue or blood samples for a fixed period of time. In addition, only people who met our inclusion criteria could conduct studies in this study. The inclusion criteria were: Patients with gastric cancer must have had their first gastric cancer surgery at the time of study and have not received any radiation or chemotherapy prior to that. This inclusion criterion further limits the number of study participants. Therefore, we may not be able to collect more samples for further verification in a short period of time, which we deeply regret and hope to get your understanding (Criteria for inclusion and exclusion of this study were added in the method tissue part).
At present, we are still collecting samples, and hope that after receiving sufficient sample sizes in the later stage, we can further explore the diagnostic and prognostic value of circRNA-0044301.
- Inhibition of circRNA-0044301 inhibited gastric tumor growth in a mouse model but the tumor still grew, albeit at a decreased rate. I think that the subcutaneous tumor model is not ideal because obviously the stomach is within the abdomen. Is this subcutaneous tumor relevant?
Answer: Yes, you are right, the best ideal model is Orthotopic transplanted tumor model[1]. However, the subcutaneous tumor model can more intuitively observe the effect of the studied gene(circRNA) on the tumor. And most nude mouse models associated with circRNA are subcutaneous tumorigenesis models[2-5]. For example, Dong-Liang Chen et al. found that knockdown of circDLG1 significantly inhibited tumor progression compared with the control treatment with their subcutaneous tumorigenesis model[6](IF=41.444). So we also constructed subcutaneous tumor in this study to observe the effect of circRNA-0044301 on gastric cancer tumor.
In addition, the role of genes on tumors is limited, so this is one of the reasons why tumor diseases have been difficult to overcome. Although circRNA-0044301studied in this study does not completely inhibit the tumor, this circRNA does have an inhibitory effect on the growth of gastric cancer tumors, which also shows that circRNA-0044301 has the potential to be used as a target for gastric cancer treatment.
We have added some discussion of animal model in our discussion part. Thanks again for your suggestion.
References:
[1]. Li, Z., et al., Circ-PTPDC1 promotes the Progression of Gastric Cancer through Sponging Mir-139-3p by Regulating ELK1 and Functions as a Prognostic Biomarker. International Journal of Biological Sciences, 2021. 17(15): p. 4285-4304.
[2]. Fang, L., et al., Circular CPM promotes chemoresistance of gastric cancer via activating PRKAA2‐mediated autophagy. Clinical and Translational Medicine, 2022. 12(1).
[3]. Peng, Y., et al., A novel protein AXIN1-295aa encoded by circAXIN1 activates the Wnt/β-catenin signaling pathway to promote gastric cancer progression. Molecular Cancer, 2021. 20(1).
[4]. Ma, X., et al., CircGSK3B promotes RORA expression and suppresses gastric cancer progression through the prevention of EZH2 trans-inhibition. Journal of Experimental & Clinical Cancer Research, 2021. 40(1).
[5]. Liu, H., et al., Circular RNA YAP1 inhibits the proliferation and invasion of gastric cancer cells by regulating the miR-367-5p/p27 Kip1 axis. Molecular Cancer, 2018. 17(1).
[6]. Chen, D., et al., The circular RNA circDLG1 promotes gastric cancer progression and anti-PD-1 resistance through the regulation of CXCL12 by sponging miR-141-3p. Molecular Cancer, 2021. 20(1).
Reviewer 2 Report
Jiang and Shen investigated expression of circRNA-0044301 in gastric cancer tissues and cell lines, and found that it was upregulated and associated with prognosis. Knockdown this circRNA could inhibit tumor growth. Mechanism investigations showed that it acted as a sponge of miRNA-188-5p, led to DAXX upregulation, supporting a key axis of circRNA-0044301/miRNA-188-5p/DAXX (ERK1/2) for GC progression. I listed below comments/suggestions to improve the manuscript:
1. About “3.4. Correlation between circRNA-0044301 and clinicopathological indicators in patients” part, I could not find Table 1.1 or 1.2. Please check. Furthermore, the authors presented that “Thus, we performed univariate Cox regression analysis using gender, age, tumor size, stage, and circRNA-0044301 expression as variables, but the results indicated that circRNA-0044301 could not be an independent prognosis marker (Table 1.2)”, it’s not precise. You should do both univariate and multivariate cox regression analyses, then you can say a molecule is an independent prognostic marker or not.
2. In Table 2, why the cutoff 1.5 was selected for longest dimension? Why not 2.0, or 5.0, please clarify. Please give a table for circRNA-0044301 expression and clinicopathological characteristics in your cohort patients.
3. Based on results of circRNA-0044301 expression levels of 6 cell lines , you could not inferred that “circRNA-0044301 is related to GC development”. You need more evidence.
4. In “3.8. Knockdown of circRNA-0044301 inhibited ERK1/2 expression and enhanced the sensitivity of cells to 5-Fu” section, the authors showed that si-circRNA-0044301 inhibited expression of ERK1/2 and p-ERK. What’s the potential mechanisms then? RIP experiments proved that ERK1/2 and p-ERK could pull down circRNA-0044301. The authors should investigate those mechanisms in detail.
5. In Fig 5, 6, 7, and 8 what do “*”, “**”, and “***” represent?
6. Fig 6A, please combine the triplicate results. Fig 7B, D, and F, and Fig 8B, etc. as well.
7. I could not find Fig 9A either. Fig 9 is suggested to be removed.
Minor point:
1. Quantitative reverse transcription polymerase reaction (qRT-PCR) should be “quantitative reverse transcription polymerase chain reaction”.
2. In Fig 1, “adjacent tumor tissues” should be “adjacent normal tissues” (in the text as well). I prefer using an uniform histogram for normal or tumor tissue, respectively.
3. Please give p value in figure legend for Fig 4.
4. The language needs polishing.
Round 2
Reviewer 1 Report
The authors have tried to address all the reviewer's concerns except for the adequate numbers of patients. They have tried to address that issue saying that the protocol is complex and it is has limited time to accrue patients
Author Response
Answer:Thank you for your understanding, it is indeed difficult to receive enough samples in a limited time, but we will try our best to continue to collect adequate samples to validate the results found in this study. Thanks again for your understanding.
Reviewer 2 Report
The authors have answered most of my comments. However, there are still a few suggestions:
1. Please double check the numbers of table and figure in the text. Such as “Table 1.1” should be “Table 1”, “Table 1.2” should be “Table 2”, “Fig 4A” should be “Fig 4”, etc.
2. I am still confused by the results of Kaplan-Meier curve and univariate analysis. The p = 0.021 in log-rank test while p = 0.531 in Cox regression analysis. Why do the p values have large difference?
3. I insist that Figure 9 should be removed, for it is worthless.
Author Response
- Please double check the numbers of table and figure in the text. Such as “Table 1.1” should be “Table 1”, “Table 1.2” should be “Table 2”, “Fig 4A” should be “Fig 4”, etc.
Answer: Thanks for your carefully checking, i have modified the numbers of these tables and Fig in our manuscript.
- I am still confused by the results of Kaplan-Meier curve and univariate analysis. The p = 0.021 in log-rank test while p = 0.531 in Cox regression analysis. Why do the p values have large difference?
Answer: Thanks for your suggestion, we checked the expression levels of circRNA-0044301 with its original results and updated some survival time of patients through telephone follow-up. Re-performed the log-rank test and the univariate cox regression analysis. The p values of them were 0.006 and 0.003, respectively. While this circRNA could not be an independent factor for its p value was more than 0.05 in the multivariate cox regression analysis. The details could be found in Table 2.
We also have to explain the different p value of log-rank test and univariate cox regression analysis. The p value of the log-rank test in this study implies the result of a comparison of the survival rates of patients in two different expression levels of this circRNA, and the survival rates of the two groups are significant different, using a non-parametric statistical method. Univariate Cox regression analysis explored whether there was a correlation between circRNA expression levels and patient survival, that is to explore whether the circRNA is an influencing factor affecting the survival process of patients. And there was no comparison of survival rates, using a semiparametric method. So there may be some difference in the results of the two methods.
- I insist that Figure 9 should be removed, for it is worthless.
Answer: Ok, i have removed Fig 9 in our manuscript. Thanks for your suggestion.